# Autophagy in Rat Müller Glial Cells Is Modulated by the Sirtuin 4/AMPK/mTOR Pathway and Induces Apoptosis under Oxidative Stress

**DOI:** 10.3390/cells11172645

**Published:** 2022-08-25

**Authors:** Mengqi Qin, Zhi Xie, Ting Cao, Zhiruo Wang, Xiaoyu Zhang, Feifei Wang, Wei Wei, Ming Jin, Jingyuan Ma, Ling Zeng, Yanan Wang, Shaonan Pei, Xu Zhang

**Affiliations:** 1Jiangxi Provincial Key Laboratory for Ophthalmology, Jiangxi Clinical Research Center of Ophthalmic Disease, Affiliated Eye Hospital of Nanchang University, Nanchang 330006, China; 2Queen Mary School, Nanchang University, Nanchang 330006, China

**Keywords:** Müller glial cell, autophagy, apoptosis, mitochondrial function, SIRT4

## Abstract

Müller glial cells (MGCs) are a group of glial cells in the retina that provide essential support to retinal neurons; however, the understanding of MGC apoptosis and autophagy remains limited. This study was aimed at investigating the role of autophagy in MGCs under normal and oxidative conditions, and identifying the underlying mechanisms. In addition, the sirtuin 4 (SIRT4)-mediated signaling pathway was observed to regulate the autophagic process in MGCs. To assess the effect of autophagy on MGC mitochondrial function and survival, we treated rMC-1 cells—rat-derived Müller glial cells—with rapamycin and 3-methyladenine (3-MA), and found that MGC death was not induced by such treatment, while autophagic dysfunction could increase MGC apoptosis under oxidative stress, as reflected by the expression level of cleaved caspase 3 and PI staining. In addition, the downregulation of autophagy by 3-MA could influence the morphology of the mitochondrial network structure, the mitochondrial membrane potential, and generation of reactive oxygen species (ROS) under oxidative stress. Moreover, SIRT4 depletion enhanced autophagosome formation, as verified by an increase in the LC3 II/I ratio and a decrease in the expression of SQSTM1/p62, and vice versa. The inhibition of AMPK phosphorylation by compound C could reverse these changes in LC3 II/I and SQSTM1/p62 caused by SIRT4 knockdown. Our research concludes that MGCs can endure autophagic dysfunction in the absence of oxidative stress, while the downregulation of autophagy can cause MGCs to become more sensitized to oxidative stress. Simultaneous exposure to oxidative stress and autophagic dysfunction in MGCs can result in a pronounced impairment of cell survival. Mechanically, SIRT4 depletion can activate the autophagic process in MGCs by regulating the AMPK–mTOR signaling pathway.

## 1. Introduction

Retinal degeneration, the main cause of blindness, can be triggered by inner environmental disturbances and can impose large economic burdens on individuals and societies. Müller glial cells, the predominant glial cells in the retina, span the whole retinal layer and contribute to the retinal structure, as well as to homeostasis, by forming an anatomical and functional link. The glutamine generated by Müller glial cells is responsible for synaptic transmission [1]. Dysfunctional Müller glial cells (MGCs) have been found in many retinopathies, including glaucoma, diabetic retinopathy, and aging-related macular disease (AMD). The selective destruction of MGCs has been found to lead to retinal dysplasia, secondary retinal degeneration, and abnormal retinal function [2,3]. Independent of their functions mentioned above, MGCs are remarkably resilient to damage and can restore lost sight by regenerating the retina in some species, such as zebrafish [4,5,6]. Furthermore, under certain circumstances, Müller glial cell lines derived from humans and rodents possess neural stem cell potential and can regenerate neurons and glia in vivo and in vitro [7,8]. This evidence indicates that Müller glial cells play an irreplaceable role in the retina under healthy or diseased conditions.

Autophagy, which has been regarded as a potential therapeutic target for multiple neurodegenerative disorders, is a finely orchestrated pathway between anabolism and catabolism that has attracted much research attention due to its cytoprotective role in physiological and pathological processes [9]. Macroautophagy, the first identified and best described form, is a highly evolutionarily conserved process of the engulfment of cytoplasmic material in a double-membraned structure with degradation upon fusion with the lysosome [10]. There is some evidence that autophagic dysfunction is involved in the onset and development of many degenerative ocular diseases, by disturbing metabolic homeostasis and inducing apoptosis. For example, in a study conducted by Deng et al. using a rhesus monkey model of chronic glaucoma, autophagic vacuole accumulation was observed in the retinal ganglion cell layer and the inner plexiform layer, and the level of autophagy-related protein was increased [11]. Retina pigmented epithelium (RPE) exposed to oxidative stress can exhibit autophagic dysfunction, and enhancing the autophagic flux can effectively defend the RPE against oxidative stress [12]. In addition, the induction of autophagy in astrocytes and microglia can induce gliosis, which has been confirmed to play a detrimental role in neurodegeneration [13]. These findings underscore the involvement of autophagy in the pathophysiological mechanism of glaucoma. Although autophagic dysregulation in vision loss has long been investigated, the effect of this critical catabolic pathway on the function of MGCs had remained an open question until now.

Sirtuins serve as nicotinamide adenine dinucleotide (NAD)-dependent histone deacetylases that are capable of regulating antioxidant and redox signaling events [14]. Sirtuin 4 (SIRT4) is a member of the sirtuin family located in the mitochondrial matrix in various organs, including the kidney, liver, heart, and brain [15]. Unlike other mitochondrial sirtuins, which can affect their targets through NAD+-dependent deacetylation, SIRT4 mainly exhibits NAD+-dependent ADP-ribosylation activity and negatively regulates glutamine metabolism and fatty acid oxidation [16]. Due to its role in lipid and glutamine metabolism, accumulated research has linked SIRT4 to the onset and development of many diseases, including diabetes; hence, this molecule has gained particular attention in recent years as a disease-specific biomarker [15]. Although its role in tissues such as the liver and kidney has been explored, the expression and activity of this molecule in the retina, specifically in MGCs, remain to be determined.

The present research provides us with a more profound understanding of the complex interplay between oxidative stress and autophagy and its relation to MGC survival and mitochondrial function, by determining a signaling mechanism underlying the regulation of autophagic function. We hypothesized that the autophagic function in MGCs was important for defense against oxidative stress, and that MGCs with activated autophagy can exert enhanced protective effects on other retinal components, such as retinal ganglion cells. Our findings demonstrated that, in MGCs, SIRT4 could act as an ATP activator whose depletion could activate the AMPK/mTOR axis and hence participate in the autophagic process. These findings extend our knowledge of the effect of autophagy on MGCs and present a wide range of possibilities for the further development of therapeutic targets for retinopathy.

## 2. Materials and Methods

### 2.1. Animals

Sprague–Dawley (SD) rats and C57BL/6 mice aged 6–8 weeks were purchased from Jiangxi University of Traditional Chinese Medicine (Nanchang, China) (license number: SCXX12018-0003) and Hunan Slac Jinda Laboratory Animal Co., Ltd. (Yueyang, China) (license number: SCXK1201-0004), respectively. The eight-week-old male wild-type mice and male SIRT4^−/−^ mice (KOCMP-75387-SIRT4-B6N-VA) (C57BL/6) used in the experiments were purchased from Cyagen Biosciences, Suzhou, China. The mice were maintained under a controlled temperature (22–24 °C) and illumination (12 h dark/light cycle) with free access to water and food. All the animals were treated in compliance with the Guide for the Care and Use of Laboratory Animals and the ARVO Statement for Use of Animals in Ophthalmic and Vision Research.

### 2.2. Transient Retinal I/R Model and Drug Administration

The retinal model was adopted, as previously described [17]. Under anesthesia (400 mg/kg chloral hydrate), unilateral transient retinal ischemia was established in the rats’ eyes. Then, 0.5% Alcaine eye drops were administered topically to anesthetize the corneas, and 1% tropicamide was used to dilate the pupils. The anterior chambers were cannulated using a 30-gauge needle with a balanced salt solution. The intraocular pressure (IOP) in one eye was maintained above the systolic blood pressure (~110 mm Hg) for 60 min, and the contralateral eye was cannulated and maintained at normal IOP. Observed blanching of the retina indicated completion of retinal ischemia. At the end of treatment, the rats were sacrificed three days after the retinal ischemia. Rapamycin (J&K Scientific Ltd., Beijing, China) diluted in dimethylsulfoxide (25 mg/mL) was dissolved in incubation medium (5% polyethylene glycol 400 and 5% Tween 80). The control group was treated intraperitoneally with sterile saline. In the treated group, rapamycin was injected intraperitoneally once a day for two days before ischemia/reperfusion (I/R) surgery and then every other day until the endpoint of each experiment [17,18].

### 2.3. Cell Culture

The experiments were performed using the rat Müller glial cell line (r-MC) (derived from lines at Northwestern University Medical School). Cells were seeded at 1 × 10^6^ per well and cultured in high-glucose Dulbecco’s modified Eagle medium (DMEM, Gibco, Australia) containing 4.5 g/L glucose, 1 mM pyruvate, and 3.97 mM L-glutamine, (Gibco by Life Technologies, Carlsbad, CA, USA) supplemented with 10% fetal bovine serum (Biological Industries, Beit-Haemek, Israel), 100 U/mL penicillin, and 100 mg/mL streptomycin at 37 °C in an atmosphere of 5% CO_2_. Oxidative stress was induced solely with hydrogen peroxide (H_2_O_2_; #323381, Sigma-Aldrich, St. Louis, MO, USA) at a concentration of 50 μM. When the cells reached 70% confluence, they were subjected to the described treatments for 24 h. For pharmacological detection, rapamycin (2 μg/mL in DMEM) (#948477, J&K Scientific, Ltd., Beijing, China), 3-methyladenine (3-MA) (1 mM/mL in DMEM) (S2767, Selleck, Houston, TX, USA), and compound C (10 μg/mL in DMEM) (S7306, Selleck, Houston, TX, USA) were administered to the r-MCs for 24 h.

### 2.4. Cell Coculture

In this experiment, Müller glial cells (MGCs) were pretreated with or without rapamycin for 24 h in six-well plates. Then, cell slides coated with retinal ganglion cells (RGCs) were placed into the six-well plates containing MGCs, and the cocultures were maintained in a 37 °C humidified atmosphere with 10% CO_2_ for 24 h [19]. Oxidative stress was induced using hydrogen peroxide (H_2_O_2_; #323381, Sigma-Aldrich, St. Louis, MO, USA) at a concentration of 100 μM [20].

### 2.5. Transfection of Lentivirus Vectors

The lentivirus expressing RNAi specific to the sirtuin 4 (SIRT4) gene, and the negative-control construct (control RNAi) were obtained from GeneChem (Shanghai, China). The target cells were plated in a 24-well plate at 40–50% confluence. On the day of infection, the appropriate titer of viral supernatant, LV-SIRT4, and control RNAi (1 × 10^8^ TU/mL) were added to the culture medium. Then, 1*HiTransG P (20 μL) (GeneChem, Shanghai, China) was added to the r-MCs (5 × 10^4^) for incubation at 37 °C for 12 h. Afterwards, the viral supernatant was replaced with fresh medium. After three days of transfection, the shRNA knockdown efficiency was determined using a fluorescence microscope (Axio Observe 3, Carl Zeiss, Germany). The following experiments were performed when the percentage of fluorescent cells reached 90%.

### 2.6. Construction and Transfection of Plasmid

The GV362 vector and SIRT4 construct were obtained from GeneChem (Shanghai, China), along with the vector element sequence of CMV–MCS–3FLAG–IRES–EGFP–SV40–Neomycin. For the SIRT4 construct, gene-specific primers for SIRT4 were designed: forward primer: 5′-TACCGGACTCAGATCTCGAGCGCCACCATGAAGATGAGCTTTGCGTTGAC-3′; reverse primer: 5′-TCCTTGTAGTCCATGGATCCGCATGGGTCTATCAAAGGCAGC-3′. RT-PCR was performed using a human RNA library as a template. The SIRT4 cDNA was then ligated into the GV362 vector between the XhoI and BamHI restriction sites. The purified fragments were cloned into the vector GV362/SIRT4 and sequenced. Transfection was performed when the seeded cells reached 70–90% confluence. Lipofectamine^TM^ 3000 was diluted with DMEM. The reagent was mixed well in Opti-MEMTM medium (2 tubes). A master mix of DNA was prepared by diluting DNA in Opti-MEMTM Medium, then adding P3000TM reagent and mixing well. Diluted DNA was added to each tube of diluted Lipofectamine 3000 reagent (1:1 ratio), and incubated for 10–15 min at room temperature. DNA–lipid complexes were added to the cells, and were incubated for two to four days at 37 °C.

### 2.7. Western Blot Analysis

Retinal tissues and r-MCs were lysed in RIPA buffer containing PMSF and phosphatase inhibitor (Solarbio, China). The protein concentration was measured using a Bradford assay (Beyotime Institute of Biotechnology) according to the manufacturer’s instructions. Protein samples (7 or 15 µg) were separated in 7.5–12% SDS–polyacrylamide gels by electrophoresis at 70 V for 2 h. The proteins were transferred to PVDF membranes using the wet transfer method. The blots were blocked in 5% skimmed milk in Tris-buffered saline/Tween-20 for 1 h, then incubated with the appropriate primary antibodies, followed by incubation with peroxidase-conjugated secondary antibodies (CST) at 1:5000. The following antibodies were used: OPA1 (#612606, BD Biosciences) at 1:1000, DRP1 (#8570, CST) at 1:1000, cleaved caspase 3 (#9664, CST) at 1:1000, SIRT4 (ab10140, Abcam) (#69786, CST) at 1:1000, LC3 (#4599, CST) at 1:1000, SQSTM1/p62 (#16177, CST) at 1:1000, p-mTOR (#5536, CST) at 1:1000, p-AMPK (#4184, CST) at 1:1000, GADPH (HC301-01, TransGen Biotech) at 1:2500, and β-tubulin (HC101-01, TransGen Biotech) at 1:2500. Finally, proteins on membranes were detected.

### 2.8. Immunofluorescence Staining

The eyeball sections were prepared as previous described [21]. The prepared tissue sections were permeated and blocked in phosphate buffered saline (PBS) containing 0.1% Triton X-100 (PBSTX) and donkey serum for 1 h at room temperature. The tissue sections and cell slides were incubated with specific primary antibodies diluted in 5% bovine serum albumin (BSA) in PBS overnight at 4 °C. Sections incubated with PBS without primary antibodies were used as negative controls. The tissue sections were incubated with secondary antibodies including donkey anti-rabbit AlexaFluor^®^ 488/594, donkey anti-mouse Alexa Fluor^®^ 488/594, and donkey anti-goat AlexaFluor 488^®^ (Abcam, Cambridge, MA, USA), diluted to 1:200 in PBS plus 0.2% Triton X-100 at room temperature for 1 h. Then, sections were counterstained with 4′,6-diamidino-2-phenylindole (DAPI; Boster, Wuhan, China). Coverslips were affixed to glass slides using anti-fading buffer (Bioworld, St. Louis Park, MN, USA) and visually examined under a Zeiss microscope (LSM800, Gottingen, Germany) equipped with epifluorescence. Digitized images were obtained by using a Zeiss camera and were processed and compiled using Photoshop. Staining was repeated three or more times for each antibody.

### 2.9. Cell Viability Assay

rMC-1 cells were seeded onto 96-well plates and incubated at a density of 1 × 10^4^ cells/well for 24 h. The cells were treated with hydrogen peroxide (H_2_O_2_; #323381, Sigma-Aldrich, St. Louis, MO, USA) at 0, 50, 100, 200, and 500 µM. The cell viability was detected with a Cell Counting Kit 8 (CCK8) (Dojindo, Japan); rMC-1 cells were treated with CCK8 at 37 °C for 1–4 h, and the absorbance at 450 nm was measured using a microplate reader (Thermo MK3, Thermo Fisher Scientific) to quantify the formazan products.

### 2.10. TUNEL Staining

Retinal cell apoptosis was detected using the TUNEL Apoptosis Detection Kit (Alexa Fluor 488) (40307ES20, Yeason). Frozen sections were prepared as mentioned above. The slides were immersed in 4% paraformaldehyde solution (dissolved in PBS) for fixation, and incubated at room temperature for 30 min. Proteinase K solution (20 μg/mL) was added dropwise to each sample for 10 min at room temperature. Terminal deoxynucleotidyl transferase (TdT) incubation buffer was added to the tissue sections. The tissue sections were observed under a confocal microscope at 20× magnification (Axio--Imager LSM-800, Zeiss, Germany). ImageJ was used to determine the number of apoptotic cells in the ganglion cell layer (GCL). The percentage of TUNEL-positive cells was calculated using the formula: number of TUNEL-positive cells/total number of cells in the GCL.

### 2.11. ROS Detection

The production of intracellular reactive oxygen species (ROS) was measured using a dihydroethidium (DHE) probe (50102ES02 Yeason). First, rMC-1 cells that had reached 90% confluence were incubated with 5 μM dihydroethidium for 15 min. Then, the cells were washed three times with PBS and collected for detection. Red fluorescence was determined using a fluorescence microscope (Axio Observe 3, Carl Zeiss, Germany) with 594 nm excitation wavelength and 525 nm emission wavelength. All the procedures were performed away from light sources. Fluorescent intensity was quantified using ImageJ.

### 2.12. Dead/Live Cell Staining

Dead/live cell staining was performed using a calcein-AM/PI kit (40747ES76, Yeasen). When the confluence of r-MCs reached 90%, they were rinsed with PBS, and staining reagent (100 μL) was added for incubation for 20–30 min at 37 °C. Then, the cells were viewed under a fluorescence microscope (Axio Observe 3, Carl Zeiss, Germany). The percentage of cell death was calculated by: % of cell death = [propidium iodide-positive cells]/[calcein-AM-positive cells +propidium iodide-positive cells] × 100%.

### 2.13. Mitochondrial Membrane Potential

The mitochondrial membrane potential was detected using a JC-10 Mitochondrial Membrane Potential Assay Kit (40752ES60, Yeasen). Briefly, cells in different groups were seeded on six-well plates (90% confluence) and incubated with 1 mL of JC-10 dyeing solution at 37 °C for 20 min. Then, the cells were washed twice with JC-10 dyeing buffer (1×). The green (excitation 490 nm, emission 530 nm) and red (excitation 525 nm, emission 590 nm) fluorescence intensities were assessed using a flow cytometer (DxFLEX, Beckman). Mitochondrial membrane potential in each group was calculated as the JC-10 green/red fluorescence intensity ratio.

### 2.14. Mitochondrial Morphology Assessment

To evaluate the mitochondrial network morphology, r-MCs were transfected with mitochondrion-targeted DsRed. The transfection was performed as described above when the seeded cells had reached 70–90% confluence. The plasmid was expressed for at least 24 h before confocal microscopy and follow-up experiments. For quantitative analysis, the fragmented mitochondrial network was photographed and ImageJ was used to skeletonize the mitochondrial images. Then, we determined the mean network size and mean rod and branch length using the Mitochondrial Network Analysis (MiNA) toolset [22].

### 2.15. ATP Content

The r-MCs transfected with LV-SIRT4 and LV-CON were harvested in PBS (0.01 M, pH 7.4) and centrifuged at 1000× *g* for 10 min at 4 °C before collection to measure the ATP content. To detect the ATP content, an ATP colorimetric assay kit (E-BC-K157-M, elabscience) was used according to the manufacturer’s instructions. All the reactions were performed in a 96-well plate by mixing 10 μL lysates with 90 μL reaction buffer. The OD value of each sample was measured at 636 nm.

### 2.16. DNA Extraction and Quantitative PCR Assay

An Ezup column animal genomic DNA extraction kit (Sangon Biotech, Shanghai, China’s B518251) was used to extract total DNA from mice tails [21]. To genotype mice, the following primers were used: for the WT allele, 5′-ACGCTACCAACCTAATGGCATC-3′ (forward) and 5′-TCCAGACACCTTGAGTCGCCTAG-3′ (reverse); for the KO allele, 5′-AC-GCTACCAACCTAATGGCATC-3′ and 5′-GAAGGCGACACAGCTACTCCATC-3′. With the use of the 2xSpecificTMTaq Master Mix (E010; Novoprotein Scientific Inc. Summit, NJ, USA), primers were selected to amplify the DNA. Initial denaturation was performed at 94 °C for three minutes, followed by 35 cycles at 94 °C for 30 s, 60 °C for 35 s, and 72 °C for 35 s, then finally an extension step at 72 °C for five minutes. All of the data from the linear amplification phase were selected. The amplified DNA was electrophoresed on a 0.8% agarose gel (with ethidium bromide) and then imaged under UV light; the image was analyzed using Image Lab. The DNA band sizes were benchmarked using a DNA ladder (DM033; Novoprotein Scientific Inc., Summit, NJ, USA).

### 2.17. Statistics

Statistical analysis was performed using graphing and statistical software (GraphPad Prism 9, GraphPad Software), and P-values less than 0.05 were considered significant. The sample size of each experiment (*n*) was determined either from triplicates or higher numbers of repetitions. All the data were expressed as the means ± SEMs, and the differences between conditions were analyzed using ANOVA followed by post hoc Tukey’s testing.

## 3. Results

### 3.1. Regulation of Autophagy Can Modulate Morphologic Alterations and Apoptosis of Müller Glial Cells under Oxidative Stress

First, we wanted to determine the subtoxic range for H_2_O_2_, and found that Müller glial cells (MGCs) exposed to H_2_O_2_ at 50 μM alone for 24 h did not show a significant loss of cell viability (Appendix A), but did show transient oxidative damage demonstrated by reactive oxidative species (ROS) generation (Appendix A). We next determined the optimal concentrations of rapamycin and 3-methyladenine (3-MA), and found that autophagy could be effectively inhibited with 3-MA at 1 mM (Appendix A), while rapamycin could significantly enhance autophagy at 2 μM (Appendix A); these results were consistent with previous research [23].

Then, we explored the role of autophagy in cells exposed to oxidative stress. Under normal conditions, the morphology of rat Müller cells (r-MCs) showed no significant differences and the cells remained dispersed, whereas MGCs exposed to H_2_O_2_ started to form an extended morphology. Moreover, MGCs subjected to H_2_O_2_ and 3-MA simultaneously became more prone to displaying bundle-like aggregations and forming large streams of cells (Figure 1A).

To determine whether autophagic dysfunction would result in r-MC toxicity, we performed immunoblot analysis of cleaved caspase 3, which is the crucial effector of apoptosis [24]. Under normal conditions, neither upregulation nor downregulation of autophagy led to any obvious change in the expression of cleaved caspase 3 compared with the control group; however, compared to the 3-MA-treated group, the rapamycin-treated group showed attenuated expression of cleaved caspase 3, with this phenomenon indicating crosstalk between autophagy and apoptosis (Figure 1B). We detected a clear trend: r-MCs subjected to 3-MA and H_2_O_2_ co-administration showed higher cleaved caspase 3 expression compared to cells exposed to H_2_O_2_ alone (Figure 1C). Consistent with the dead/live cell staining of r-MCs under different conditions (Figure 1D,E), only a negligible number of cells in the control group were observed to be PI-positive. r-MCs exposed to H_2_O_2_ alone showed increased numbers of PI-stained cells, whereas the majority of r-MCs co-treated with H_2_O_2_ and 3-MA were PI-positive. These results suggest that the inhibition of autophagy could increase the susceptibility of r-MCs to oxidative stress, and that autophagic dysfunction under sublethal oxidative stress can lead to obvious r-MC death.

### 3.2. Regulation of Autophagy Can Impact Mitochondrial Function of r-MCs under Oxidative Stress

To further investigate the underlying mechanism of autophagy-induced apoptosis, we focused on the mitochondrial function of r-MCs in response to oxidative stress and autophagic modulation. Mitochondrial dynamics, a determinant of mitochondrial quality, are related to mitochondrial function [25]. In order to assess mitochondrial fusion and fission under normal conditions and oxidative stress, fluorescence of mitochondrion-targeted DsRed was used. Upon either upregulation or downregulation of autophagic function, string-like mitochondria could be observed in normoxic r-MCs, while under oxidative stress the morphology of mitochondria clearly became clumped and fragmented with the downregulation of autophagy (Figure 2A). Then, we used the Mitochondrial Network Analysis toolset (MiNA) to quantify the mitochondrial morphology [22]. The treatment of MGCs with H_2_O_2_ or 3-MA alone did not significantly affect the mitochondrial network structure, while simultaneous exposure to oxidative stress and autophagic dysfunction resulted in a pronounced impairment of mitochondrial morphology, as indicated by a much greater number of fragmented mitochondrial individuals, shorter mean branch per network length, and decreased numbers of branches per network (Figure 2B–D). These results illustrate that autophagic dysfunction under oxidative stress could lead to a reduction in mitochondrial network structure. To investigate in detail the molecular manipulation of mitochondrial dynamics, we analyzed the expression levels of optic atrophy gene 1 (OPA1), a mitochondrial fusion protein, and dynamin-related protein 1 (DRP1), a mitochondrial fission protein, after rapamycin or 3-MA treatment [26]. As depicted in Figure 2E, compared with normal controls, OPA1 levels were downregulated by approximately 40% in 3-MA-treated r-MCs, but cells treated with rapamycin did not show a significant change. Significant changes were not detected in the immunoblot analysis of DRP1 (Figure 2F). Meanwhile, under oxidative stress, 3-MA could damage r-MCs by inhibiting OPA1 expression; hence, the mitochondrial damage caused by oxidative stress was exaggerated by inhibiting mitochondrial fusion (Figure 2G). Treatment with rapamycin exerted a protective effect on r-MCs by inhibiting the expression of DRP1, phosphorylation of which can lead to apoptotic cell death (Figure 2H) [26].

The mitochondrial membrane potential (ΔΨm) is an independent indicator of mitochondrial quality, responsible for basic functions of the mitochondria, such as protein import and synthesis in addition to ATP generation [27,28]. The sustained depolarization of mitochondria can induce apoptosis in target cells [29]. As depicted in Figure 3A, the mitochondrial membrane potential was monitored by JC-1 fluorescent staining. The H_2_O_2_ and 3-MA co-treatment of r-MCs decreased the ΔΨm, indicating that the dysfunction of autophagy could promote the stress-induced dissipation of ΔΨm (Figure 3A,B).

Reactive oxygen species (ROS), production of which mainly occurs during mitochondrial respiration, can be detrimental to cells by inducing excessive oxidative stress [30,31]. To evaluate the role of autophagy in this process, we analyzed ROS levels after treatment with rapamycin or 3-MA under oxidative stress. Here, compared with cells incubated with H_2_O_2_ alone, the upregulation of autophagy under oxidative stress did not show an effect on ROS generation, while a significant increase in the ROS generation of r-MCs could be detected following 3-MA and H_2_O_2_ co-administration (Figure 3C,D).

Taking these results together, we can conclude that the downregulation of autophagy leads to disordered mitochondrial dynamics and excessive ROS generation, especially under oxidative conditions. The data reveal that autophagic dysfunction predisposed cells to oxidative stress by influencing mitochondrial quality and apoptosis. The downregulation of autophagy resulted in greater susceptibility to H_2_O_2_ treatment.

### 3.3. Activation of Autophagy by Rapamycin Can Help Retinal Neurons to Survive under Ischemia/Reperfusion Injury and Oxidative Stress

Müller glial cells are among the first cells to respond to pathogenic insult in the retina. Rapamycin is a classic autophagic activator. In order to verify whether the autophagic pathway played a crucial role in neurodegenerative retinal disease, rats administered with rapamycin or otherwise were sacrificed three days after ischemia-reperfusion injury to detect the effect of autophagy on gliosis and retinal cell death [17].

The gliosis of MGCs can lead to retinal neurodegeneration, characterized by cellular hypertrophy and upregulated glial fibrillary acidic protein (GFAP) [21]. The immunofluorescence results showed that GFAP expression was increased and reorganized after retinal ischemia-reperfusion injury, indicating reactive MGC gliosis, while rapamycin treatment could clearly mitigate MGC gliosis (Figure 4A).

To investigate the effect of rapamycin on retinal ganglion cell apoptosis, we performed TUNEL staining on the retina and found that three days after ischemia-reperfusion injury there was a large amount of retinal ganglion cell apoptosis, and that rapamycin treatment could alleviate cell death. These results indicate that the activation of autophagy can effectively protect the retina by inhibiting MGC gliosis and retinal cell apoptosis induced by external injuries (Figure 4B).

To determine whether the induction of autophagy in Müller cells was sufficient to support retinal neuronal survival, r-MCs were cultured in a well containing a cell slide coated with rat-derived retinal ganglion cell line R28. By co-immunostaining for Brn3a, a classic retinal ganglion cell (RGC) marker [32], and for cleaved caspase 3, we found a significant impact on the survival of RGCs grown with Müller cells pretreated with rapamycin for 24 h, compared with RGCs cultured with untreated MGCs after exposure to 100 μM H_2_O_2_ (Figure 4C). All of these results demonstrate that the activation of autophagy in the Müller glia significantly inhibited RGC death under stress conditions.

### 3.4. Depletion of Sirtuin 4 Can Increase Autophagic Function In Vitro and In Vivo

Sirtuin 4 (SIRT4) is involved in many catabolic cellular processes, including autophagy. In this research, we aimed to determine whether and how SIRT4 could affect the autophagic process in MGCs. We previously showed that SIRT4 is highly expressed in MGCs; in the current study we performed SIRT4 and glutamine synthetase (GS) co-staining to verify that SIRT4 was highly expressed in MGCs, which was consistent with our previous analysis (Figure 5) [30]. To determine whether the impact of SIRT4 depletion was associated with autophagy, LC3 was used as a specific marker for autophagosomes. In the in vivo experiment, we found that the retinas of SIRT4-knockout mice showed lower LC3 II/I ratios and higher SQSTM1/p62 expression, compared with wild-type mice (Figure 6A–C). The ectopic expression of SIRT4 resulted in a profile of downregulated autophagy (Figure 6D–F). Then, we downregulated SIRT4 expression by transfecting LV-SIRT4 RNAi into MGCs. An increased LC3 II/I ratio accompanied by a decreased expression of SQSTM1/p62 could be observed in V-SIRT4 r-MCs, compared with V-CON r-MCs (Figure 6G–I). To verify our conclusion, we visualized autophagic puncta by immunostaining LC3 on r-MCs. Consistent with the immunoblot analysis, the number of LC3B dots per cell was approximately twofold higher in V-SIRT4 compared with V-CON r-MCs, as determined by confocal imaging analysis (Figure 6J,K). The in vitro and in vivo experimental results showed that SIRT4 can inhibit the formation of MGC autophagosomes.

### 3.5. SIRT4 Participates in Modulating r-MC Autophagic Function through AMPK/mTOR Signaling Pathway

After determining that SIRT4 might be responsible for MGC autophagy, we decided to decipher the signaling pathway monitored by SIRT4. The immunoblot analysis results showed that SIRT4 depletion in r-MCs activated AMPK phosphorylation and inhibited mTOR phosphorylation (Figure 7A,B). It has been reported that the AMPK/mTOR axis is a classical signaling pathway for the manipulation of autophagy, given that ATP is a potent upstream activator of the axis. To further investigate the connection between ATP homeostasis and SIRT4-mediated autophagy regulation, cells with or without SIRT4 depletion were examined, and it was observed that the absence of SIRT4 led to decreased ATP generation (Figure 7C). Then, we incubated compound C (10 μM) [33] with V-CON and V-SIRT4 r-MCs. As shown in Figure 6D–F, only the downregulation of SIRT4 expression led to a significant increase in autophagy, while lower LC3 II/I and higher SQSTM1/p62 could be observed in V-SIRT4 r-MCs where AMPK phosphorylation was inhibited using compound C.

## 4. Discussion

Neurodegeneration, which occurs with age, is characterized by neuronal injury and loss [34]. Accumulating research has linked the onset and progression of neurodegenerative diseases to dysfunctional autophagy. Similarly, glaucoma is characterized by neurodegeneration of the retina, an extension of the brain [35], and can also be induced by abnormal autophagic function [36,37,38]. However, the impact of autophagy on Müller glial cells (MGCs), the main glial cells in the retina, remains unclear. Overall, in the present study, we illustrated the following: 1. Dysregulated autophagy can make MGCs more prone to mitochondrial dysfunction and apoptosis under oxidative stress; simultaneous autophagic dysfunction and exposure to H_2_O_2_ synergistically impair MGC survival and mitochondrial function. 2. The activation of autophagy can strengthen the protective effect of MGCs on RGC apoptosis. 3. Sirtuin 4 (SIRT4) depletion can upregulate autophagosome formation in MGCs through the AMPK/mTOR axis, and might exert an impact on autophagic function in this manner.

In this research, we found that autophagic dysfunction sensitized MGCs to oxidative stress through a mitochondrion-mediated process, manifested by disordered mitochondrial dynamics, increased ROS generation, and a depolarized mitochondrial membrane potential. The in vivo experiments and cell coculture experiment verified that the upregulation of autophagic function could protect the ischemic retina. Next, by focusing on sirtuins, we investigated the causes of autophagy in MGCs. We demonstrated that SIRT4 depletion can activate autophagic function in MGCs. Moreover, by inhibiting AMPK using compound C, we demonstrated that SIRT4 can activate the mTOR signaling pathway by repressing AMPK activation (Figure 8).

Autophagy is characterized by a lysosomal degradative process, and autophagic dysfunction has been found to correlate with several neurodegenerative diseases [39]. Systemic autophagy deficiency in vivo did not lead to significant alterations in histology and visual function in ATG4B-knockout mice, but could modulate the capability of retinal ganglion cells (RGCs) to respond to optic nerve axotomy [40]. Similar results were found in Beclin-1^+/−^ mice, in which the retinal morphology and photoreceptor function did not show significant alterations under normal lighting conditions, although there was retinal degeneration after 2 h of exposure to bright light [41]. All of these results indicate that autophagy is a cytoprotective mechanism helping cells to cope with various injurious conditions.

Our results in this research showed that, like other retinal components, MGCs demonstrated strong resistance to dysregulated autophagy under normal conditions, and that the downregulation of autophagic function or exposure to mild oxidative stress alone did not have a significant effect on MGC morphology or apoptosis. To more clearly demonstrate the beneficial effect of autophagy on MGCs under oxidative stress, which is one of the leading causes of retinal degeneration, we exposed rat Müller cells (r-MCs) to H_2_O_2_ with modulation of autophagy by a pharmacological approach. Interestingly, we found that autophagic dysfunction could render MGCs vulnerable to oxidative stress. All of these findings indicate that MGCs possess the potential ability to autoregulate to defend themselves against autophagic dysfunction as well as oxidative stress. However, simultaneous exposure to oxidative stress alongside autophagic dysfunction in MGCs can synergistically impair cell survival. After H_2_O_2_ exposure and the downregulation of autophagic function, apart from the phenotype that showed deterioration in MGC apoptosis during oxidative stress, we also observed glial aggregation, which has been reported to lead to retinal disorganization and can only be found in severely degenerated retinas [42]. These data corroborate the common opinion that an appropriate level of autophagy is essential to protect MGCs and maintain the retinal architecture after oxidative injury.

In order to understand how autophagic dysfunction induces apoptosis, we next examined mitochondrial functions. Mitochondria are highly dynamic organelles that act as energy reservoirs, participating in cellular redox as well as bioenergetic regulation. However, in addition to playing pivotal roles in biosynthesis and cellular energetics, mitochondrial components are also key operators that regulate cell death programming. Normal mitochondrial function is maintained by continuous fusion and fission. It is widely accepted that mitochondrial fusion can protect mitochondria from various insults by diluting mtDNA damage [43], while fission can accelerate the progression of apoptosis by participating in Bax-mediated permeabilization in the outer mitochondrial membrane [43,44]. Fragmented mitochondria are commonly found in apoptotic cells, along with cytochrome c which refines the cristae of healthy mitochondria and is released when mitochondrial outer-membrane pores are formed [45]. Dysfunctional mitochondria can lead to excessive generation of reactive oxygen species (ROS), which in turn exerts a progressive deteriorating effect on mitochondrial DNA [46]. Excessive ROS generation can accelerate the loss of mitochondrial membrane potential, leading to cytochrome c release and caspase 3 activation [47].

Previous research revealed that 24 h of H_2_O_2_ exposure could induce reduced-branching mitochondrial formation in a human Müller glial cell line, but did not exert any effect on mitochondrial copy numbers [48]. In the present research, compared with normal conditions, MGCs with downregulated autophagy showed obviously fragmented mitochondria under oxidative stress, a phenomenon that might be induced by the downregulation of mitochondrial fusion protein OPA1. Under oxidative stress, the downregulation of autophagy could induce ROS generation and mitochondrial membrane potential (MMP) depolarization, and we suspected that autophagic dysfunction killed the cells via mitochondrial apoptosis. These results suggest that autophagic dysfunction can be considered responsible for triggering disturbances in mitochondrial redox homeostasis and mitochondrial dynamics, hence inducing mitochondrion-mediated apoptosis in r-MCs under oxidative stress.

One of the particular findings of our research is that the induction of autophagy specifically in Müller cells is important to support neuronal survival. Firstly, we found that the upregulation of autophagic function by rapamycin effectively alleviated the MGC gliosis and retinal ganglion cell apoptosis induced by ischemia-reperfusion injury, confirming the protective effect of the autophagic function. Moreover, for Müller cells pretreated with rapamycin for 24 h, an increase in RGC survival could be observed; this finding supports our hypothesis that the upregulation of autophagy in MGCs can effectively enhance the MGCs’ ability to protect RGCs from oxidative injury. These data confirm a potential role of Müller cells with enhanced autophagy in the prevention of RGC apoptosis.

SIRT4 is one of the sirtuins contained within mitochondria, and its presence or absence has been found to be correlated with many metabolism-related diseases and processes, including cancer, neurodegeneration, and aging [15]. As a NAD^+^-dependent enzyme that participates in fatty acid and glutamine metabolism, there is no doubt that SIRT4 is crucial in influencing the rate of the utilization and the metabolic fates of lipids and carbohydrates [16]. It has been found that SIRT4 can regulate ATP levels by interacting with adenine nucleotide translocator 2 (ANT2) [49]. Previous studies carried out by our group demonstrated that SIRT4 co-localized with MGCs and participated in regulating MGCs and the MGC marker glutamine synthetase (GS) [21]. It has been well established that AMPK is an energy sensor that can regulate energy homeostasis by partly enhancing autophagic function [50]. Combined with the evidence that SIRT4 can inhibit AMPK by regulating ATP homeostasis, this suggests that the SIRT4–AMPK axis is important for connecting the metabolic energetic state to cellular catabolic or anabolic processes, and may thus determine the induction of pathways including autophagy [49,51,52]. The autophagic regulation pattern of SIRT4 is dependent on cell type and pathophysiological state; for example, Shaw et al. demonstrated that SIRT4 can inhibit cellular autophagy in hepatocytes by regulating glutamine utilization, and can hence activate TORC1 [52]. Wang et al. found thatSIRT4 overexpression in hepatocellular carcinomas could activate autophagy by inhibiting glutamine metabolism and increasing ADP/AMP levels, thereby activating the LKB1/AMPKα/mTOR signaling pathway [16]. In the cardiovascular system, SIRT4 overexpression has been reported to protect cardiomyocytes from doxorubicin-induced cardiotoxicity by inhibiting Akt/mTOR-dependent autophagy [53]. In the pancreas, SIRT4 can deactivate autophagic function by deregulating AMPK and directly inhibiting insulin secretion [54]. Interestingly, we also found that under 50μM H_2_O_2_, the expression level of SIRT4 showed a tendency to decrease, while the phosphorylation of AMPK increased (Appendix A). Combined with previous findings, we postulated that MGC can activate the AMPK-mediated molecular mechanism by inhibiting SIRT4 expression to defend itself against the presence of hydroperoxide. However, the underlying mechanism accounting for their correlation needs further study.

There remain several limitations to this current study. First, although our results potentially demonstrate that SIRT4 depletion can enhance autophagy, it remains to be determined whether SIRT4 can indirectly affect cells’ vulnerability to oxidative stress by modulating autophagy. Second, in the present research, we only detected the expression under SIRT4 depletion of autophagy players and not autophagic functions themselves; to further explore SIRT4′s effect on autophagic function, more extensive studies are required to explain how SIRT4 impacts autophagy.

In summary, our research detected for the first time the functional changes in MGCs after modulating autophagic function under oxidative or normal conditions. Our results suggest that the disruption of autophagy does not have a significant effect on the death or mitochondrial functions of MGCs, which are among the most resilient cell components in the retina. However, dysfunctional apoptosis can render MGCs more susceptible to oxidative stress. We suspect that autophagic dysfunction kills cells by mitochondrial apoptosis. We also found that the SIRT4/AMPK axis participated in regulating the autophagic process, which indicates promising potential for SIRT4 in retinal neuroprotection, as an important factor linking metabolic energic states to cellular catabolic and anabolic processes.

## Figures and Tables

**Figure 1 cells-11-02645-f001:**
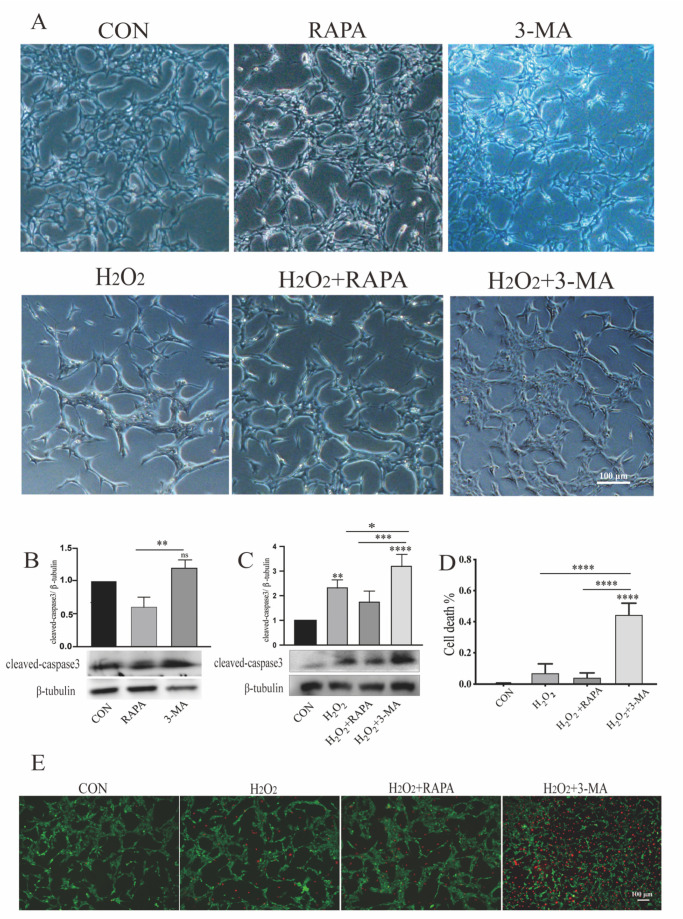
Regulation of autophagy can modulate the morphologic alterations and apoptosis of MGCs under oxidative stress. r-MCs were treated with rapamycin or 3-MA under normal conditions or oxidative stress for 24 h. (**A**) Bright-field microscopy was used to detect the morphologic alterations of MGCs. r-MCs treated with H_2_O_2_ and 3-MA formed aggregates. Scale bar: 100 μm. (**B**,**C**) Apoptosis of MGCs was detected by changes in the expression of cleaved caspase 3 by western blotting. Appropriate downregulation of autophagy induced by 3-MA (1 mM) was not toxic to r-MCs under normoxia, while it could increase vulnerability to oxidative stress. Data are shown as means ± SEMs (*n* = 3 per group; * *p* < 0.05, ** *p* < 0.01, *** *p* < 0.001 and **** *p* < 0.0001). (**D**) Cell death in different groups was quantified. Data are shown as means ± SEMs (*n* = 3 per group; **** *p* < 0.0001). (**E**) Live/dead cell staining for oxidative stress and autophagic dysfunction showed obvious MGC apoptosis. Scale bar: 100 μm. RAPA, rapamycin; 3-MA, 3-methyladenine.

**Figure 2 cells-11-02645-f002:**
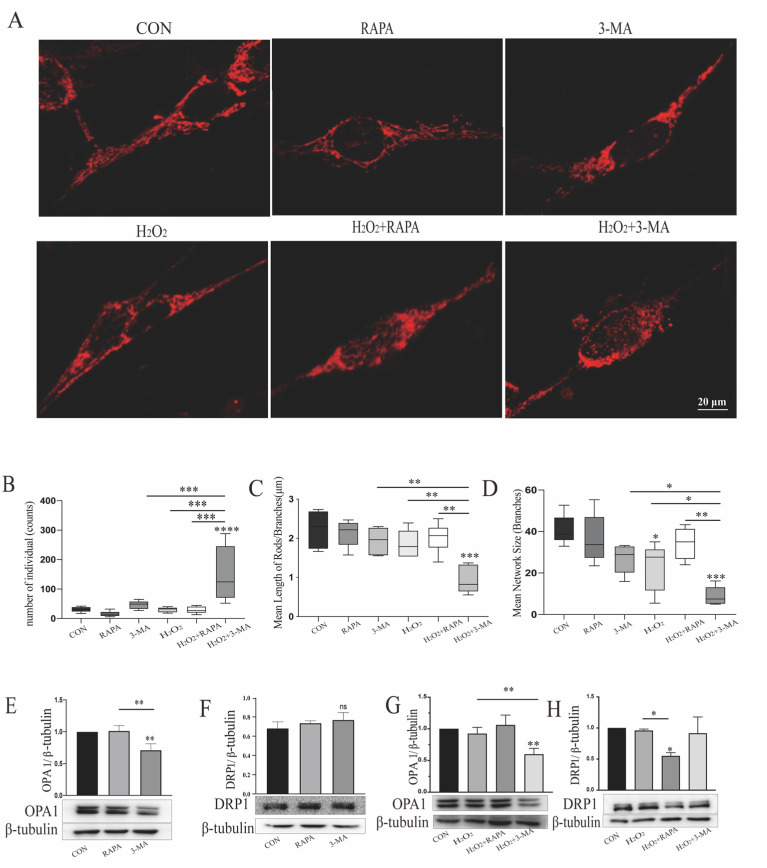
Regulation of autophagy can impact the mitochondrial morphology of MGCs under oxidative stress. (**A**) Fluorescence of mitochondrion-targeted DsRed was used to detect mitochondrial morphology. r-MCs treated with 3-MA under oxidative stress showed fragmented mitochondria. Scale bar: 20 μm. (**B**–**D**) MiNA was used to analyze the related indices of the mitochondrial network structure. Data are shown as means ± SEMs (*n* = 5–7 per group; * *p* < 0.05, ** *p* < 0.01, *** *p* < 0.001 and **** *p* < 0.0001). (**E**–**H**) Quantitative analysis of OPA1 and DRP1 was performed. Data are shown as means ± SEMs (*n* = 3 per group; * *p* < 0.05 and ** *p* < 0.01). RAPA, rapamycin; 3-MA, 3-methyladenine; OPA1, optic atrophy gene 1; DRP1, dynamin-related protein 1.

**Figure 3 cells-11-02645-f003:**
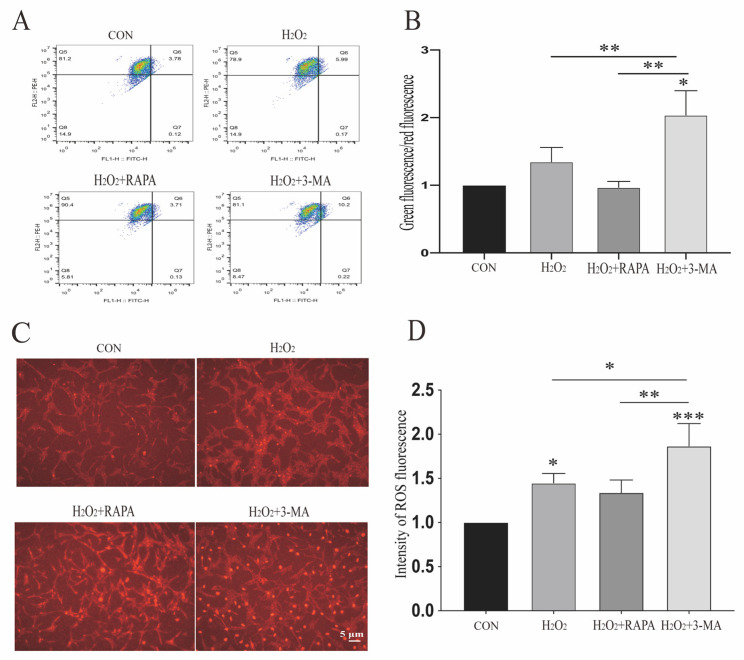
Regulation of autophagy can impact mitochondrial membrane potential and ROS generation of MGCs under oxidative stress. (**A**) In conditions of oxidative stress, the mitochondrial membrane potential significantly decreased after treatment with 3-MA (1 mM). (**B**) Quantitative analysis of mitochondrial membrane potential in different groups was performed. Data are shown as means ± SEMs (*n* = 3 per group; * *p* < 0.05 and ** *p* < 0.01). (**C**) ROS generation in r-MCs with upregulation and downregulation of autophagy under hypoxia was detected by immunofluorescence using a DHE kit. Scale bar: 5 μm. (**D**) Quantitative analysis of ROS fluorescent intensity in different groups was performed. Data are shown as means ± SEMs (*n* = 3 per group; * *p* < 0.05, ** *p* < 0.01 and *** *p* < 0.001). RAPA, rapamycin; 3-MA, 3-methyladenine; OPA1, optic atrophy gene 1; DRP1, dynamin-related protein 1.

**Figure 4 cells-11-02645-f004:**
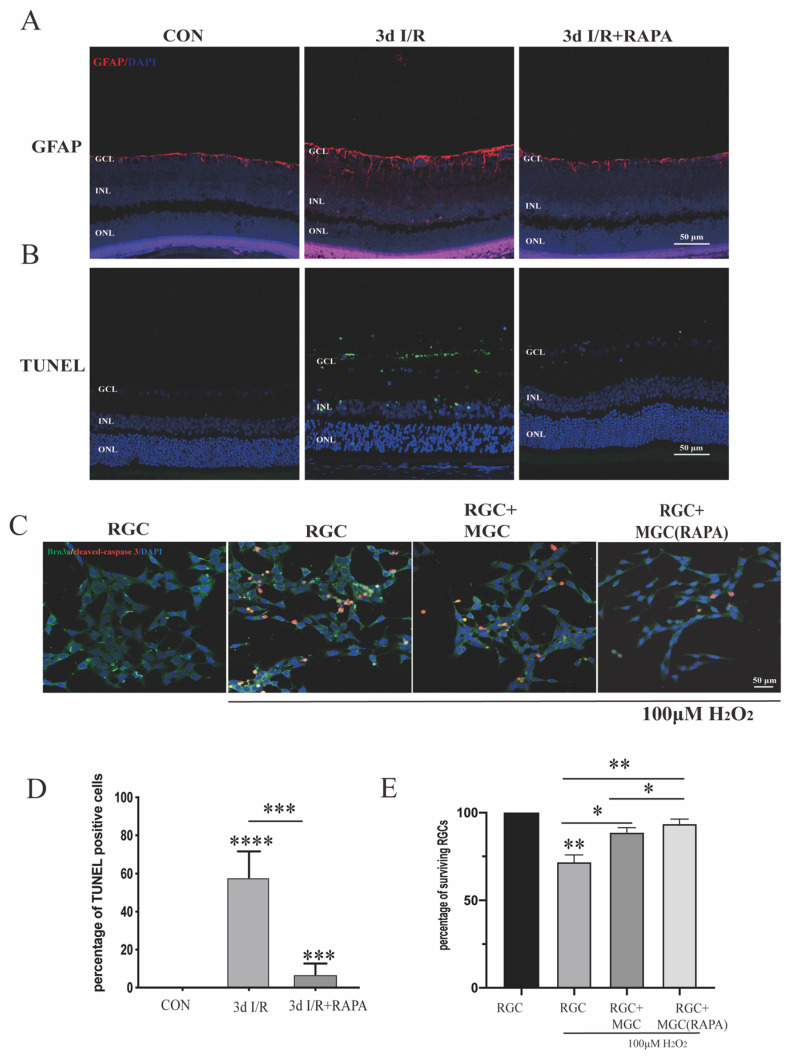
Rapamycin treatment can alleviate MGC gliosis and retinal cell apoptosis. (**A**) GFAP immunostaining in rat retina, in control, ischemia-reperfusion, and ischemia-reperfusion plus rapamycin groups (scale bar: 50 μm). (**B**) TUNEL staining in rat retina, in control, ischemia-reperfusion, and ischemia-reperfusion plus rapamycin groups (scale bar: 50 μm). (**C**) Brn3a and cleaved caspase 3 co-immunostaining was performed on the r-MC cell line cocultured with R28 (scale bar: 50 μm). (**D**) Quantitative analysis of TUNEL staining in rat retina. Data are shown as means ± SEMs (*n* = 3 per group; *** *p* < 0.001 and **** *p* < 0.0001). (**E**) Quantitative analysis of survival of RGCs. Data are shown as means ± SEMs (*n* = 3 per group; * *p* < 0.05 and ** *p* < 0.01). GCL, ganglion cell layer; INL, inner nuclear layer; ONL, outer nuclear layer; I/R, ischemia/reperfusion; RAPA, rapamycin.

**Figure 5 cells-11-02645-f005:**
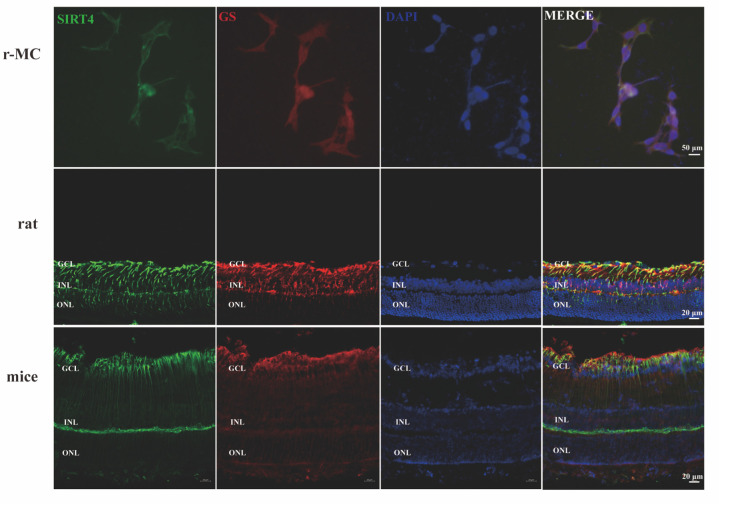
SIRT4 is highly expressed on MGCs. SIRT4 and GS co-immunostaining was performed on the r-MC cell line (scale bar: 50 μm), rat retina (scale bar: 20 μm), and mouse retina (scale bar: 20 μm). GS, glutamine synthetase; GCL, ganglion cell layer; INL, inner nuclear layer; ONL, outer nuclear layer.

**Figure 6 cells-11-02645-f006:**
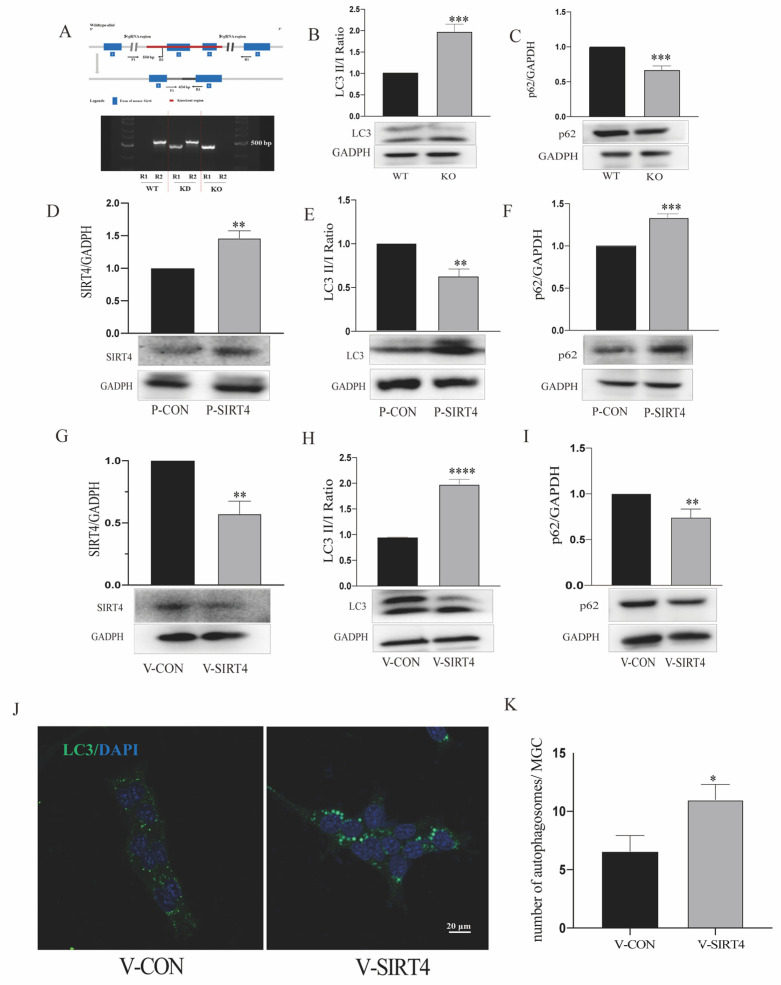
SIRT4 can modulate autophagic function in vivo and in vitro. (**A**) Genotyping identification of SIRT4-knockout mice. (**B**,**C**) Autophagic flux in retinas of wild-type and SIRT4-knockout mice was monitored by LC3-II/I conversion with anti-LC3 and SQSTM1/p62 antibodies. (**D**) The overexpression efficiency of the SIRT4 plasmid was validated by western blotting. (**E**,**F**) Autophagic flux in P-CON and P-SIRT4 r-MCs was monitored by LC3-II/I conversion with anti-LC3 and SQSTM1/p62 antibodies. Data are shown as means ± SEMs. (**G**) The knockout efficiency of the SIRT4 lentivirus was validated by western blotting. (**H**,**I**) Autophagic flux in V-CON and V-SIRT4 r-MCs was monitored by LC3-II/I conversion with anti-LC3 and SQSTM1/p62 antibodies. Data are shown as means ± SEMs. (**J**) Autophagic puncta in MGCs were obtained by immunostaining with an LC3 antibody (scale bar: 20 μm). (**K**) Quantitative analysis of autophagic puncta in MGCs. Data are shown as means ± SEMs. (*n* = 3 per group; * *p* < 0.05, ** *p* < 0.01, *** *p* < 0.001, and **** *p* < 0.0001). SIRT4, sirtuin 4; WT, wild type; KO, knockout; p62, SQSTM1/p62.

**Figure 7 cells-11-02645-f007:**
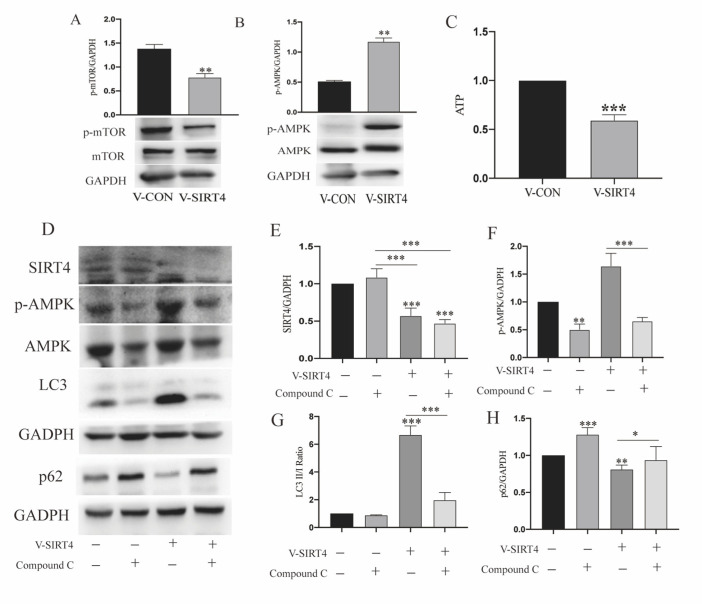
SIRT4 participates in modulating r-MC autophagic function through AMPK/mTOR signaling pathway. (**A**,**B**) Protein expression levels of p-AMPK and p-mTOR in whole-cell lysates from cells transfected with V-SIRT4 (*n* = 4 per group; ** *p* < 0.01). (**C**) ATP generation of MGCs transfected with V-SIRT4. Data are shown as means ± SEMs (*n* = 3 per group; *** *p* < 0.001). (**D**–**H**) Protein expression levels of SIRT4, p-AMPK, p-mTOR, LC3, and SQSTM1/p62 in whole-cell lysates from cells transfected with V-SIRT4 and treated with compound C (n = 3 per group; * *p* < 0.05, ** *p* < 0.01, and *** *p* < 0.001). p62, SQSTM1/p62.

**Figure 8 cells-11-02645-f008:**
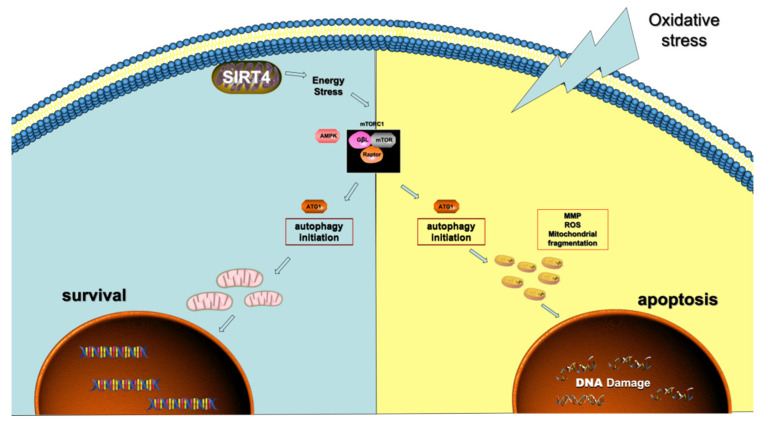
Schematic representation of the AMPK/mTOR signaling pathway regulated by SIRT4 in MGCs. Under normal conditions, SIRT4 depletion in MGCs inhibits ATP generation in mitochondria, which impinges on p-AMPK modulation, thus activating mTOR-mediated autophagy. Under oxidative stress, dysfunctional autophagy can induce disturbed mitochondrial dynamics by increasing mitochondrial fission and decreasing fusion, leading to MGC apoptosis.

## Data Availability

The original contributions presented in the study are included in the article/Appendix A, and further inquiries can be directed to the corresponding author.

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
