# Peer review of "Autophagy in Rat Müller Glial Cells Is Modulated by the Sirtuin 4/AMPK/mTOR Pathway and Induces Apoptosis under Oxidative Stress"

_cells, 2022, doi:10.3390/cells11172645_

Round 1

Reviewer 1 Report

The manscript by Qin et al. concerns an important topic about the role of autophagy in Müller glial cells to prevent retinal neurodegeneration. The authors demonstrate that autophagy stimulation by rapamycin, prevents cell death in a Müller glial cell line, but also in vivo in rats subjected to retinal ischemia/reperfusion. They also demostrate that in cells and mice where SIRT4 is deleted autophagy is increased in an AMPK dependent way. Results are interesting but unfortunately the paper is badly written and needs a careful revision of the English language before it can be considered. There are notable mistakes in the methods section which need to be carefully addressed and some data need further support.

The manuscript is not acceptable for publication in the present version. It must be revised, rewritten and if authors are interested, submitted again.

Comments

Introduction

1. P62 please change to SQSTM1/p62 according to Klionsky guide

Methods

This section has many mistakes and some information is lacking in several parts.

1. The surgery procedure for ischemia is not well described and the experimental conditions for in vitro experiments are not clearly described. It is not clear whether cells were exposed to hydrogen peroxide solely or to hydrogen peroxide plus starvation.

2. The method for monitoring mitochondrial morphological changes (DsRed) is not described.

3. Some sections have important mistakes:To monitor ROS production the authors describe that dihydroethidium (DHE) was used, which is clearly what the used, but in the methods section they say that dichlorofluorescence was monitored, as if they used dichlorofluorescein diacetate to measure ROS.

4. In legends to Fig. 1 and Supplemental Fig. 1 they describe that DHE immunofluorescence detection kit was used for ROS. This is a mistake since DHE is a compound that when oxidized in the presence of ROS, becomes fluorescent, it is not an antibody.

5. According to the methods section Hoechst was used to monitor living cells. This is not possible, Hoechst is a fluorochrome that marks nuclei, either from dead or living cells. Clearly, they used calcein-AM and propidium iodide.

6. In some parts, the methods are written as a laboratory protocol, and not formally for publication.

7. It is not clear whether ATP content or ATP synthase activity was measured. In Figure 6 D ATP levels are reported, buy according to the methods section it was ATP synthase activity what was measured.

8. The authors use “normoxic” and “hyperoxic” to refer to control and H2O2 exposed cells, which is not correct because cells are not exposed to a change in oxygen concentration, but to oxidative stress by H2O2 exposure.

RESULTS

1. Some of the results need quantitative data in order to be reinforced: number of ROS producing cells, mitochondrial morphological changes (number of fragmented mitochondria, circularity, ferret diameter), number of dead cells (TUNEL-positive cells).

2. Some images need to be improved:

Fig 1 A 3-MA panel is too dark; Fig 1D the H2O2+ 3-MA panel is too dark. Fig. 1 B active caspase-3, the rapamycin line is overexposed

Fig 2 A images are too small and changes in mitochondrial morphology are not visible.

3. To study mitochondrial dynamics the authors monitored the changes in the total content of OPA and DRP1, which is not sufficient, particularly because phosphorylated Drp1 was not investigated, and because morphological changes were not quantified. The images are too small to distinguish the presumed changes.

4. The most striking results are those in cells where SIRT4 was deleted or those in SIRT4 KO mice, which demostrate that SRT4 deletion increases autophagy. However, it was not tested whether this cells/mice were less vulnerable to hydrogen peroxide exposure or ischemia, which would have made an important contribution.

Discussion

The authors demostrate that autophagy induction by rapamycin is important to prevent retinal cell death in mice and the death of Müller cells in culture. However, these experiments do not demostrate that the induction of autophagy specifically in Müller cells, is important to support neuronal survival. It would be interesting to investigate in the rat model whether the protective effect of rapamycin against ischemia in ganglion cells is due to the induction of autophagy either in Müller glial cells or in ganglion cells. An immunocytochemistry against LC3 of p62 in combination with markers of Müller cells (glutamine synthetase) and neurons (NeuN) can give some information. In SIRT4 KO mice, it will be very interesting to study whether autophagy is stimulated in Müller cells.

Reviewer 2 Report

Dear authors,

Your work shows great interest in the field, being the muller cells an important target for vision problems. However, I found some details that need to be corrected in order to fully complete this interesting work:

1.- Abbreviations need to be explained once in each big section, it is not readable otherwise.

2.- It is really important to explain where the culture cells are used (in vitro) against the in vivo ones.

3.- In vitro experiments: to test the autophagy activation or inactivation with the different drugs, it is necessary to perform autophagy flux experiments ( read how here PMID: 35087090; 34233024; 33634751)

4.- I would use another more quantifiable ROS measure, i.e. Amplex red, also it is important to be careful if you use a mitochondrial probe, it can be altered by the membrane potential changes.

5.-How are you sure that the effects in vivo with the Rapamycin IP are true? Is it published that it works? I would test some rapamycin targets in vivo.

6.- Figure 6: you should show total proteins, not only phosphorylated ones. When applicable comment 3 (autophagy flux needs to be measured)

7.- Show bigger images in Figure 2A and Figure 5C.

Reviewer 3 Report

Major

1.       In Figure 1A, the cell numbers or density in each of the images are quite differences. For example, from the image in Figure 1A, the cell density in RAPA-treated cell image is lower than in the control cell image. Does this mean the RAPA could kill the MGCs? I would like to suggest authors to use the Apoptosis flow cytometry kit to analyze the apoptosis level of the drug treatments as in Figure 1A. This could give a clearer picture for the readers.

2.       In Figure 1C, is the changes between H2O2 and H2O2-RAPA significant? If it is not, will it mean that RAPA does not have effect or protective effect on oxidative stress in MGCs here?

3.       Same as above, in Figure 1D, could authors use a statistic approach to show this staining result instead of images only? From the images of H2O2 and H2O2+RAPA, the differences of staining are not so clear cut.

4.       In Figure 2B, the expression of OPA1 was downregulated with treatment of 3-MA, while in Figure 2D, firstly, is there any significantly difference between H2O2 and H2O2+3-MA? Secondly, not significantly difference between H2O2 and H2O2+RAPA, is that mean RAPA or autophagy could not affect the OPA1 expression in oxidative stress condition?

5.       In Figure 2G, similar as above. Could this insignificancy due to the H2O2 itself? Is it possible that H2O2 the authors used is not fresh enough as the differences of 1/ OPA1 (Figure 2D), 2/ DRP1 (Figure 2E) and 3/ Green Fluorescence ratio (Figure 2G) are not significant when compare with untreated controls in all tests?

6.       Could authors show the SIRT4 expression levels in 1/ V-CON and V-SIRT4 cells, 2/ P-CON and P-SIRT4 cells, and 3/ WT and KO retinal tissues, respectively? These are important information for readers that SIRT4 is clearly downregulated in the KO mice, RNAi-treated cells and SIRT4 plasmid construct-transfected cells, when compare with their controls, respectively.

7.       On Line 467, author stated that “SIRT4 depletion can upregulate autophagic function”. Authors only showed the autophagy players expression regulation, but not functional changes. Some functional studies could help on this statement. For example, authors could perform Electron Microscopy to identify autophagic structures changes or testing the Fusion with the Lysosome.

Minor

1.       Could authors also show the bar chart of the other protein targets in Figure 6 (SIRT4, p-AMPK, Tublin)?

2.       Authors treated the cells with Compound C (Dorsomorphin) which is inhibitor of pAMPK. Could authors give the information of this chemical in method part, for example, brand name?

3.       On the Line 404, authors stated that “Consistent with the immunoblot analysis, the number of LC3B dots per cell was also approximately two-fold higher in V-CON r-MC vs V-SIRT4 r-MC as determined by confocal imaging analysis”. According to this sentence, authors observed that LC3B dots per cell number is 2-fold higher in V-control cells when compare with V-SIRT4 cells, is that correct?

Round 2

Reviewer 1 Report

The authors have adequately addressed most of the comments of this reviewer. They have improved the manuscript and figures and made additional co-culture experiments that support their hypothesis. They showed that autophagy induction in Müller cells under oxidative stress improves retinal ganglion cells survival. This is an important contribution to the field. Authors have corrected the Methods Section, which is now presented more clearly.

The present version of the manuscript is suited for publication although there are some details in figures and methods that should be addressed before it can be accepted.

Major comments

Results

·      Page 4 Lines 108-109. No depolarization of MGCs is shown in Fig. S1B as it is described in the text. Please correct.

·      Fig. 1D. What do the authors mean by “Cell death %”. Please explain in the methods section how the percent of cell death was calculated.

·      Fig. 1E is not described in Legend to Fig. 1

·      Fig. 3B Why do the authors report the green fluorescence/total fluorescence ratio in mitochondrial membrane potential determinations? Normally the green fluorescence (monomeric form)/ red fluorescence (aggregated JC-10) ratio is reported. The wave lengths used for JC-10 determinations should be indicated in the graph and the Methods section. Detailed information for these determinations is needed in methods.

·      Fig. 3D What does ROS level mean? Mean fluorescence? number of ethidium-positive cells?. Please indicate this in the figure and the Methods section.

Discussion section

The authors show that autophagy induction in Müller cells in normal conditions

Do the authors checked whether the abundance of SIRT4 is altered in oxidative stress conditions? Can they include a comment on that?

Page 13 lines 397-398

 The authors conclude: “These results suggest that autophagic dysfunction can be considered a process responsible for triggering disturbances in mitochondrial redox homeostasis and mitochondrial dynamics, hence inducing mitochondrion-mediated apoptosis, in r-MCs.”

I basically agree with this conclusion but it may be emphasized that this phenomenon occurs under oxidative stress conditions, and not in normal conditions.

Minor comments

Abstract

LC3-I/LC-3 II should be LC3II/LC3-I

P62 should be p62 in all figures

Page 13 line 92

MGCs with downregulated autophagy showed obviously fragmented mitochondria under hypoxia. Is hypoxia correct?

Author Response

Reviewer#1

The authors have adequately addressed most of the comments of this reviewer. They have improved the manuscript and figures and made additional co-culture experiments that support their hypothesis. They showed that autophagy induction in Müller cells under oxidative stress improves retinal ganglion cells survival. This is an important contribution to the field. Authors have corrected the Methods Section, which is now presented more clearly.

The present version of the manuscript is suited for publication although there are some details in figures and methods that should be addressed before it can be accepted.

Response: We thank the Reviewer#1 for the positive comments and careful review which helped improve the manuscript. Based on your comments, we performed the following changes and the changes have been marked in red in the revised manuscript:

Major comments

Results

Page 4 Lines 108-109. No depolarization of MGCs is shown in Fig. S1B as it is described in the text. Please correct.

Response: Sorry for our mistake and we have corrected it. (Page 3 Lines 108-109 in the revised manuscript)

Fig. 1D. What do the authors mean by “Cell death %”. Please explain in the methods section how the percent of cell death was calculated.

Response: Cell death% was determined by the percentage of propidium iodide-positive cells from the total number of cells (calcein-AM-positive cells plus propidium iodide-positive cells). And we have explained it in the Methods section. (Page 17 Lines 586-588 in the revised manuscript)

Fig. 1E is not described in Legend to Fig. 1

Response: Thank you for noticing this, and we have added figure legend about Figure 1E. (Page 4 Lines 140-141 in the revised manuscript)

Fig. 3B Why do the authors report the green fluorescence/total fluorescence ratio in mitochondrial membrane potential determinations? Normally the green fluorescence (monomeric form)/ red fluorescence (aggregated JC-10) ratio is reported. The wave lengths used for JC-10 determinations should be indicated in the graph and the Methods section. Detailed information for these determinations is needed in methods.

Response: Thank you for pointing these out, we have redetermined mitochondrial membrane potential by calculating the green / red fluorescence intensity ratio, and added detailed information in the Methods section (Page 17 Lines 593-597 in the revised manuscript).

Fig. 3D What does ROS level mean? Mean fluorescence? number of ethidium-positive cells?. Please indicate this in the figure and the Methods section.

Response: ROS level indicates the Intensity of ROS fluorescence, and we have re-named the Y axes in the figure and added relevant information Methods section. (Page 17 Lines 595-596 in the revised manuscript)

Discussion section

The authors show that autophagy induction in Müller cells in normal conditions

Do the authors checked whether the abundance of SIRT4 is altered in oxidative stress conditions? Can they include a comment on that?

Response: Thank you for your suggestion.

Compared to the control group, the expressional level of SIRT4 is downregulated and the p-AMPK is activated under 50μM H2O2 in MGC. Base on this finding, we postulated that that MGC can activate the AMPK-mediated molecular mechanism by inhibiting SIRT4 expression to defend itself against the the certain amounts of hydroperoxide. However, the underlying mechanism about their correlation needs further study. This part of discussion was added in Discussion Section in Page 14, Lines 427-432. We believe this finding can be used as a reference for the researchers who aim to investigate underlying mechanisms of pathophysiological characteristics in muller glia cell.

Page 13 lines 397-398

 The authors conclude: “These results suggest that autophagic dysfunction can be considered a process responsible for triggering disturbances in mitochondrial redox homeostasis and mitochondrial dynamunfics, hence inducing mitochondrion-mediated apoptosis, in r-MCs.”

I basically agree with this conclusion but it may be emphasized that this phenomenon occurs under oxidative stress conditions, and not in normal conditions.

Response: Thank you for your reminding, and we have changed relevant description in the conclusion. (Page 13 Lines 400-403 in the revised manuscript)

Minor comments

Abstract

LC3-I/LC-3 II should be LC3II/LC3-I

Response: Thank you for your suggestion, “LC3 I/II” has been corrected to “LC3 II/I”

P62 should be p62 in all figures

Response: Thank you for your suggestion, “P62”has been corrected to “p62” in all figures.

Page 13 line 392

MGCs with downregulated autophagy showed obviously fragmented mitochondria under hypoxia. Is hypoxia correct?

Response: Thank you for your reminding, and the“hypoxia” has been corrected to “oxidative stress”. (Page 13 Lines 393-396 in the revised manuscript)

Reviewer 2 Report

Dear authors,

After your revisions, the quality of the work has improved a lot, congratulations! I have just a few comments to do. One is that you are right about the Amplex red, but I took it as an example, there are other methods to measure ROS, other probes which react with superoxides, for example, but the quantification improved a lot the information provided. As you don't have "enough time" to probe properly the autophagic activity (I imagine as an editorial request), I would tone down a bit the allegations about autophagy,. Finally, about the rapamycin IP treatment, although I still find it very striking, I checked your reference, but I would include some others, to make that treatment more credible.

Author Response

Reviewer#2

Dear authors,

After your revisions, the quality of the work has improved a lot, congratulations! I have just a few comments to do. One is that you are right about the Amplex red, but I took it as an example, there are other methods to measure ROS, other probes which react with superoxides, for example, but the quantification improved a lot the information provided. As you don't have "enough time" to probe properly the autophagic activity (I imagine as an editorial request), I would tone down a bit the allegations about autophagy,. Finally, about the rapamycin IP treatment, although I still find it very striking, I checked your reference, but I would include some others, to make that treatment more credible.

Response: We appreciate the Reviewer#2's comments concerning our manuscript. Those comments are all valuable and very helpful for revising and improving our paper, as well as the important guiding significance to our researches.

As correctly mentioned by Reviewer#2, performing quantifiable ROS measure can represent more intuitive understanding about ROS generation in different groups. In this research, we used DHE, a common method to detect ROS generation, and quantified ROS level by detecting intensity of ROS fluorescence, thus the quantitative results can be used as a reference to confirm the ROS generation in different groups (1-3). In our future work, we will try to use other optimised quantifiable method to detect ROS generation in accordance with the Reviewer#2 request.

The rapamycin IP treatment is one of the appropriate routes to attenuate retinal neuronal injury in the preclinical model (4-7). We appreciated for Reviewer#2’s suggestion and added other reference to support us that administrated rapamycin intraperitoneally can effectively protect retina from ischemia/reperfusion model (Page 15 Lines 475 in the revised manuscript). In our future animal experiments, we will try to test some rapamycin targets in the rapamycin IP treated rat or use some other rapamycin administration routes (such as subconjunctival and intravitreal injection ), to assess the protective effect of rapamycin in retinal cell apoptosis.

Reference:

  • Roehlecke C, Schumann U, Ader M, Knels L, Funk RH. Influence of blue light on photoreceptors in a live retinal explant system. Mol Vis. 2011;17:876-884.
  • Zhang L, Chen J, Yan L, He Q, Xie H, Chen M. Resveratrol Ameliorates Cardiac Remodeling in a Murine Model of Heart Failure With Preserved Ejection Fraction. Front Pharmacol. 2021;12:646240. doi:10.3389/fphar.2021.646240
  • Tadokoro KS, Rana U, Jing X, Konduri GG, Miao QR, Teng RJ. Nogo-B Receptor Modulates Pulmonary Artery Smooth Muscle Cell Function in Developing Lungs. Am J Respir Cell Mol Biol. 2016;54(6):892-900. doi:10.1165/rcmb.2015-0068OC
  • Rao YQ, Zhou YT, Zhou W, Li JK, Li B, Li J. mTORC1 Activation in Chx10-Specific Tsc1 Knockout Mice Accelerates Retina Aging and Degeneration. Oxid Med Cell Longev. 2021;2021:6715758. doi:10.1155/2021/6715758
  • Kida T, Oku H, Osuka S, Horie T, Ikeda T. Hyperglycemia-induced VEGF and ROS production in retinal cells is inhibited by the mTOR inhibitor, rapamycin. Sci Rep. 2021;11(1):1885. doi:10.1038/s41598-021-81482-3
  • Russo R, Varano GP, Adornetto A, et al. Rapamycin and fasting sustain autophagy response activated by ischemia/reperfusion injury and promote retinal ganglion cell survival. Cell Death Dis. 2018;9(10):981. doi:10.1038/s41419-018-1044-5
  • Su W, Li Z, Jia Y, Zhuo Y. Rapamycin is neuroprotective in a rat chronic hypertensive glaucoma model. PLoS One. 2014 Jun 12;9(6):e99719. doi: 10.1371/journal.pone.0099719.

Reviewer 3 Report

The authors have addressed most of the comments and the quality of the manuscript has been improved a lot. Here are a few minor comments that should be addressed:

1.      Lines 108-109, “demonstrated by depolarization and reactive oxidative species (ROS) generation (Figure S1B)”. However, there is no depolarization of MGCs is shown in the Figure S1B.

2.      No description of Figure 1E in the legend of Figure 1.

3.      In Figure 3D, could authors state the units of the ROS level on Y-axis? For example, percentage? Intensity of ROS fluorescence?

Author Response

Reviewer#3

Comments and Suggestions for Authors

The authors have addressed most of the comments and the quality of the manuscript has been improved a lot. Here are a few minor comments that should be addressed:

Response: We appreciate the reviewer's constructive evaluation of our work and thank reviewer for valuable suggestions. The changes have been marked in red in the revised manuscript

  1. Lines 108-109, “demonstrated by depolarization and reactive oxidative species (ROS) generation (Figure S1B)”. However, there is no depolarization of MGCs is shown in the Figure S1B.

Response: Sorry for our mistake and we have corrected it. (Page 3 Lines 107-108 in the revised manuscript)

  1. No description of Figure 1E in the legend of Figure 1.

Response: Thank you for noticing this, and we have added description about Figure 1E. (Page 4 Lines 140-141 in the revised manuscript)

  1. In Figure 3D, could authors state the units of the ROS level on Y-axis? For example, percentage? Intensity of ROS fluorescence?

Response: In this research, we detect ROS level by monitor the intensity of ROS fluorescence by imageJ. We have added relevant information in the figure legend and Methods section and re-named the Y axes. (Page 7 Lines 205-206, Page 17 Lines 581 in the revised manuscript)